# Development and Disorders of the Airway in Bronchopulmonary Dysplasia

**DOI:** 10.3390/children10071127

**Published:** 2023-06-29

**Authors:** Douglas Bush, Courtney Juliano, Selina Bowler, Caterina Tiozzo

**Affiliations:** 1Division of Pediatric Pulmonology, Department of Pediatrics, Mount Sinai Hospital, Icahn School of Medicine, New York, NY 10029, USA; douglas.bush@mssm.edu; 2Division of Neonatology, Department of Pediatrics, Mount Sinai Hospital, Icahn School of Medicine, New York, NY 10029, USA; courtney.juliano@mssm.edu; 3Department of Pediatrics, New York University Langone-Long Island, Mineola, NY 11501, USA; selina.bowler@nyulangone.org

**Keywords:** bronchopulmonary dysplasia, airway development, airway disorders

## Abstract

Bronchopulmonary dysplasia (BPD), a disorder characterized by arrested lung development, is a frequent cause of morbidity and mortality in premature infants. Parenchymal lung changes in BPD are relatively well-characterized and highly studied; however, there has been less emphasis placed on the role that airways disease plays in the pathophysiology of BPD. In preterm infants born between 22 and 32 weeks gestation, the conducting airways are fully formed but still immature and therefore susceptible to injury and further disruption of development. The arrest of maturation results in more compliant airways that are more susceptible to deformation and damage. Consequently, neonates with BPD are prone to developing airway pathology, particularly for patients who require intubation and positive-pressure ventilation. Airway pathology, which can be divided into large and small airways disease, results in increased respiratory morbidity in neonates with chronic lung disease of prematurity.

## 1. Overview of Airway Development

One of the key milestones of the transition from intrauterine to extrauterine life is the shift from a fluid-filled positive-pressure condition in the lung to one of negative pressure air breathing. This transition is physiologically complex and subject to a myriad of potential complicating factors. It is especially challenging in an underdeveloped respiratory system, as is present in the preterm infant. The immature respiratory system is highly susceptible to environmental insults, which can disrupt normal developmental programming. The resulting pathology, bronchopulmonary dysplasia (BPD), is characterized by varying degrees of impaired alveologenesis, pulmonary vascular abnormalities, and airway anomalies. In its current form, often referred to as the ‘new BPD’, the pathophysiology is predominantly one of arrested or impaired development, which is in contrast to the features of lung injury that were the hallmark of the ‘old BPD’ phenotype [1].

The primitive lung develops at around 4–6 weeks of human gestation from a structure called the laryngotracheal groove, a small diverticulum budding from the ventral wall of the foregut into the surrounding mesenchyme [2]. This structure gives rise to the trachea by ventral caudal elongation and, through a process of lateral septation and distal-to-proximal closure, separates from the esophagus except at the site of the future hypopharynx and larynx [3]. Subsequently, from the distal end of the primitive trachea, the two mainstem bronchi arise. The third generation of branching takes place during the sixth-week post-conception, producing 10 tertiary bronchi on the right and eight on the left, referred to as the large conducting airways in this discussion. 

Normal development of the conducting airways and eventual alveolar formation is a complex process mediated by the sequential expression of transcription factors and coordinated cross-talk between mesenchyme and epithelium (including endothelial, neuronal, and immune cells) [3,4,5]. Thyroid transcription factor-1 (TTF-1 or NKX2.1) is a critical, early-expressed epithelial transcriptional factor that is essential in the development of epithelial progenitors from the laryngotracheal groove and proximal bronchi [6]. SOX -2 and the retinoic receptor transcription factor activation pathway also play important roles in the formation of the laryngotracheal groove [7,8,9]. Branching morphogenesis distal to the trachea mainly depends on NKX2.1 and fibroblast growth factor (FGF) ligands, especially FGF10 [7,10,11]. Concomitantly, the vascular structures of the respiratory system begin to form with the pulmonary arteries developing from the sixth aortic arch and the pulmonary veins emerging from the developing heart [12]. By the end of the embryonic stage, the larynx, trachea, lung primordia, lung lobes, and the bronchopulmonary segments have formed. 

During the second phase of lung development, called the pseudoglandular phase, occurring between 7 and 17 weeks of gestation, there is ongoing branching morphogenesis that, through a continuous process of repetitive sprouting and bifurcations, leads to the formation of the pre-acinar airways accounting for approximately 20 generations of the respiratory tree [13]. These airways will be referred to as the small conducting airways in this discussion. 

In human embryonic lungs, smooth muscle cells form a sleeve surrounding the developing conducting airway epithelium. Lateral budding occurs at points of opening in the smooth muscle coat through which the lateral bud epithelium protrudes and extrudes. Lateral branching does not occur where the smooth muscle sleeve is present [14] (Figure 1).

During this phase, both respiratory epithelium and mesenchyme begin to differentiate. Basal and neuroendocrine cells appear at around 8–10 weeks of gestation and, by approximately 16 weeks of gestation, ciliated and secretory cells emerge in the proximal airways [15]. Finally, in the distal regions, cuboidal cells, the primordial type II pneumocytes, which are filled with glycogen—an essential component of surfactant—start to appear [16].

The mesoderm gives rise to intrapulmonary arteries (branching in parallel with the bronchial and bronchiolar tubules), supportive cartilage in the airways, and bronchial smooth muscle [15,17]. Development and proper patterning of the cartilage rings and smooth muscle cells are dependent on critical signaling from fibroblast growth factor 10 (FGF10), fibroblast growth factor receptor 2b (FGFR 2b), fibroblast growth factor 18 (FGF18), and sonic hedgehog, which interact with bone morphogenetic proteins to coordinate this process [18,19,20].

Cartilage begins to appear during the 4th week of gestation in the trachea, the 10th week in the main bronchi, and the 12th week in the segmental bronchi and peripherally extends until about 2 months following birth. Cartilage mass continues to increase throughout childhood [16]. 

The formation, timing, and orientation of the subsequent segmental airway branches involve continuous crosstalk between the epithelium and mesenchyme. This process is mediated by several important and evolutionarily conserved developmental signaling molecules such as FGFs [21], Hedgehog (HH) [22], WNT [23], and TGF-beta [24].

Additionally, interbranch lengths and timing of branching are thought to be partially determined by the presence of calcium waves. Calcium waves are activating peristaltic waves of smooth muscle contractions that are proposed to function as cellular clocks [25]. Inhibition of the sarcoendoplasmic reticulum calcium -ATPase (SERCA) calcium pump results in a reversible inhibition of these calcium waves (and consequently of the peristalsis) and disrupted branching morphogenesis of the airways. It is hypothesized that SERCA integrates inputs from morphogens such as FGF10, leading to differential Ca^2+^ levels at branching tips, indicating a potential “timing” for cell migration and branch formation. The waves of rhythmic peristalsis move fluids along a proximal to distal axis, down the lumen of the developing airways. The generated hydraulic forces dilate the epithelial distal tips and are responsible for the outward bulging of the epithelium at the weak points where smooth muscle cells are not present (orifice). Through this process, new epithelial branches are formed. These branches subsequently acquire a smooth muscle coat (stalk) until a point where new branches will appear [25] (Figure 1). By the end of this phase, the complete numbers of pre-acinar pulmonary vessels are present in each segment with a pattern now similar to that of an adult lung.

Branching morphogenesis is completed during the canalicular phase, which occurs between 17 and 25 weeks gestation. During this phase, the terminal bronchioles are formed, the alveolar epithelium begins to differentiate into type I and type II cells, and the mesenchyme thins, allowing for the formation of a rudimentary blood-gas interface. At the end of the canalicular phase, non-ciliated bronchiolar cells appear and surfactant production by type II alveolar epithelial cells begins [11,26,27]. Finally, there is increased vessel proliferation and development of capillary networks around the rudimentary gas exchange structures in the peripheral mesenchyme. 

By the final two stages of lung development, the saccular phase, which takes place between 24 and 36 weeks gestation, and the alveolar phase, which occurs between 36 weeks gestation and 8 years of age, airway differentiation and development are largely complete. The saccular phase is characterized by the formation of primitive terminal airspaces and the beginning of surfactant production by Type 2 pneumocytes. By the end of this stage, three additional generations of alveolar ducts have formed, and mucous and ciliated cells are developing in the conducting airways. Secondary septation begins in the alveolar phase with the formation of new secondary septae dividing the saccules into smaller alveoli with a 20-fold increase in the gas exchange area throughout the course of this developmental phase [28]. 

Each step in this sophisticated process is necessary to achieve normal lung development and normal respiratory function. Any disruption of signaling between cells may be sufficient to alter the entire process. Because the lung continues to undergo a significant portion of its development after birth, derangements of this complex developmental program—which may involve molecular, environmental, or mechanical insults—can occur pre- or post-natally. This susceptibility is particularly evident in the setting of premature birth where structural immaturity, surfactant deficiency, and frequent exposure to environmental stressors (inflammatory or oxidative) create a setup for disrupted development, often leading to the diagnosis of bronchopulmonary dysplasia (BPD), the most common complication of prematurity [29].

BPD was originally described in 1967 as a pulmonary disease affecting preterm infants following prolonged exposure to oxygen therapy and positive-pressure mechanical ventilation [30]. The classic BPD phenotype was characterized by lung injury and fibrosis. As neonatal care has become increasingly sophisticated, with a larger number of very-low-birthweight and extremely preterm infants surviving, these features have become less common and a new BPD phenotype has emerged. Key characteristics of the ‘new’ BPD reflect the pathophysiology of delayed or disrupted maturation and development [29]. Severe BPD often involves overlapping pathology of the lung parenchyma, vasculature, and airways. The remainder of this review will focus on summarizing the clinical airway phenotypes presented by premature infants affected by BPD. 

## 2. The Large Airways

The large airways, defined as the trachea (including the subglottic space through the carina), mainstem bronchi, lobar bronchi, and segmental bronchi, all form the proximal portion of the conducting airways. While the conducting airways do not provide gas exchange, an understanding of air flow in these large airways is relevant in pathophysiologic states. Small reductions in airway diameter result in significant increases in airway resistance (R = 1/r^4^) [31]. Thus, any changes to an already small conducting airway can lead to substantial increases in airway resistance in the neonatal population. 

Airway arborization occurs throughout the tracheobronchial tree with the cross-sectional area of the airways increasing in more distal lungs. The smallest cross-sectional area of the respiratory system exists at the segmental bronchi (3rd branching). It is here that bronchi are most impacted by changes in intrathoracic pressure. Airway collapse can be observed throughout the central airways when the intrathoracic pressure exceeds the intraluminal pressure with forced exhalation, coughing, or in disease states such as tracheomalacia or extrinsic compression of the airways (Table 1) [32]. 

As with older children and adults, the neonatal airway is impacted by changes in transmural pressures in the airway during breathing. With spontaneous breathing, airway caliber increases with inspiration, owing to the negative pressure exerted upon the respiratory system. With exhalation, less negative pressure (or positive pressure when coughing) leads to a reduction in the caliber of the airway (Figure 2) [34]. Infants, particularly those born prematurely, have a propensity to experience an exaggerated dynamic impact on the airways given their immature and more compliant cartilaginous support structures [35,36]. Further, the posterior trachealis muscle lining the posterior airway is not bound to a rigid support system and can protrude into the airway in both non-disease (e.g., coughing) and disease (e.g., disorders associated with hypotonia, or impaired innervation via the recurrent or superior laryngeal nerves) states [37,38]. As with cartilage, smooth muscle is immature in preterm infants and has been demonstrated to have increased muscle tension and tactile strength as gestation progresses [39,40] Increases in airway resistance require an increase in work of breathing by the infant, leading to even larger pressure swings in order to overcome the increased resistance and can exacerbate dynamic collapse [41,42]. Work of breathing in infants is further challenged by a highly compliant thoracic cage and a mechanically disadvantaged diaphragm, which can contribute to early respiratory failure [43,44].

It is important to note that positive-pressure ventilation (both invasive and non-invasive) may contribute to increased airway radius, a reduced thickness of airway cartilage and smooth muscle, and can contribute to epithelial damage [45]. As a result, there may be an increased risk of airway collapsibility and increased airway resistance in infants who have previously received positive-pressure ventilation [46].

Disorders of the large airways can be differentiated into congenital or acquired (Table 1). Congenital abnormalities are rare and unlikely to occur at any increased incidence in premature infants, but should always be considered when airway obstruction, recurrent infections, or increased work of breathing is a concern, and an evaluation of the airway anatomy (e.g., imaging or direct visualization with flexible bronchoscopy) can help guide management. Acquired lesions, such as tracheomalacia, bronchomalacia, or subglottic stenosis, are far more common in this population, although the exact prevalence is unknown [34]. 

### 2.1. Subglottic Stenosis

Acquired subglottic stenosis occurs at a relatively high frequency with 0.9–8.3% of intubated neonates experiencing stenosis [47,48]. With a narrow cricoid, infants are uniquely predisposed to subglottic injury when respiratory challenges lead to frequent or prolonged endotracheal intubations and, at times multiple or traumatic attempts at intubation [49,50]. Sub-glottic stenosis frequently presents as a fixed airflow obstruction with inspiratory stridor or bi-phasic wheezing. If severe, airway resistance will be high with increased work of breathing evident on examination. 

The assessment of subglottic stenosis requires a flexible laryngoscopy or, more often, direct laryngoscopy with rigid bronchoscopy in order to accurately diagnose and grade the lesion [51]. Mild grades of subglottic stenosis are typically well tolerated; however, more severe grades may require tracheostomy placement and eventual airway reconstructive surgery when the infant is older and the airway is larger [52]. The mainstay of subglottic stenosis management is an emphasis on prevention with the goal of limiting endotracheal intubation through the early use of nasal continuous positive airway pressure (CPAP) or, when intubation is required, to do so nasally given the preferable tube stabilization benefits with nasal intubation [34,53]. Medical therapies are limited in their efficacy, with acid-suppressive medications providing questionable benefits when gastroesophageal reflux is suspected as the source of subglottic edema, and not without their potential side effects [54]. In mild disease, endoscopic balloon dilation can be considered; however, there is a high risk of failed dilation in this population and surgical interventions may ultimately be required [55]. In severe subglottic stenosis, surgical interventions, including tracheostomy with tube placement or laryngotracheal reconstruction may be necessary.

### 2.2. Tracheomalacia and Bronchomalacia

The prevalence of tracheomalacia in BPD is not well known, but is likely occurring at a higher incidence than has previously been appreciated, with estimates as high as 48–60% in infants with severe BPD [56,57]. As defined earlier, the risk factors include immature cartilaginous support structures, laxity of the posterior trachealis muscle, and challenged respiratory mechanics, frequently necessitating increased work of breathing. In a recent retrospective study looking at risk factors for tracheomalacia in preterm infants affected by BPD, among the 58 preterm infants evaluated, 50% were diagnosed as having tracheomalacia. The presence of severe BPD, prolonged intubation, and higher peak inspiratory pressures were most closely associated with a diagnosis of tracheomalacia [58].

Currently, there is no standardized definition for tracheomalacia; however, most experts consider a dynamic collapse of more than 50% during spontaneous breathing to be abnormal [59]. Tracheomalacia can manifest from an anterior defect in the airway or through a defect, or laxity, of the posterior trachealis muscle. The abnormality can involve a short segment of the trachea, the entirety of the trachea, can extend into the mainstem bronchi (tracheobronchomalacia), or can manifest entirely within the mainstem bronchi alone (bronchomalacia). 

Signs and symptoms include respiratory distress, homophonous (constant pitch equally heard throughout the lungs) wheezing, coughing (from retained secretions), acute desaturation, and hypoventilation episodes, with agitation or apneas in neonates. While there are no specific therapies available to treat or manage tracheomalacia, non-invasive continuous positive airway pressure (CPAP) support seems to be a mainstay of therapy. The use of CPAP helps to provide an airway distending pressure, reducing airway collapse and its accompanying resistance, facilitating exhalation, and preventing hyperinflation [60]. For infants with primary defects of the posterior trachealis muscle, sympathomimetic agonism targeting smooth muscle tone has been considered; however, there is limited evidence supporting the efficacy of such therapies at this time (e.g., bethanechol) [61]. 

In general, as infants grow, the diameter of their airways enlarges. As such, tracheomalacia tends to clinically improve by the age of 2 years old in this population [62]. While medical interventions are limited to CPAP, for more severe tracheomalacia, surgical interventions may be required. Tracheotomy with a tracheostomy tube can provide both a means of delivering invasive CPAP and, at times, can be used to bypass the site of airway collapse [60]. 

External compression of the airway by cardiovascular structures occurs at a fairly low incidence in the population with BPD; however, with aberrant anatomy and small airways, even slight reductions in the airway diameter can lead to significant increases in airway resistance. Sites of external airway compression include the anterior trachea from an innominate artery or an aberrant aortic arch, the left mainstem through compression from cardiomegaly secondary to enlarged left-sided cardiac structures, or bilateral mainstem bronchi by enlarged and engorged pulmonary arteries in the setting of pulmonary hypertension or left-to-right shunts (e.g., patent ductus arteriosus). Often, vascular compression appears pulsatile on direct visualization. While management of these individuals occasionally requires non-invasive or invasive positive airway pressure, there are at times medical (treatment of the PH or cardiac disease) or surgical interventions (e.g., aortoplasty or PDA closure). 

In adults with obstructive airway diseases such as severe asthma or chronic obstructive pulmonary disease (COPD), an entity called excessive airway dynamic collapse (EADC) has been described [63]. In the adult literature, EADC refers to the effect increased expiratory effort has on airway smooth muscle as a physiologic choke point where amplified transmural forces lead to excessive collapse [64,65]. In disease states, such as BPD, where large airway abnormalities are common, the large obstructing airways may be impacting the smaller airways through various means including regional hyperinflation and stretch, compression, atelectasis, inflammation related to retained secretions, aspiration, or other small airways injury. While small airway abnormalities have been physiologically appreciated in infants, children, and adults formerly with BPD, the etiology of small airway disease remains elusive. It is not unreasonable to consider the physiologic concepts behind excessive dynamic airway collapse as an etiology for small airway disorders in this population [66].

## 3. The Small Airways 

The small airways, also known as the peripheral airways, are non-cartilaginous and begin at the 8th generation of branching. They consist of the bronchioles, terminal bronchioles, and respiratory bronchioles [13]. In preterm infants born between 22 and 32 weeks gestation, during the end of the canalicular stage and the saccular stage of lung development, the conducting airways are fully formed and the formation of the respiratory bronchioles is nearly complete. However, these structures are still immature and therefore susceptible to injury and further disruption of development [67]. Classic BPD, predominantly defined by autopsy findings, was noted to involve pathologic findings of metaplasia of bronchial smooth muscle, bronchial necrosis, mucosal metaplasia, and excessive airway secretions [30]. These findings are not typical of the new BPD [68]; however, some structural abnormalities of the small airways such as peribronchial fibrosis and mucous plugging do occur, which can lead to fixed airway narrowing and increased airway resistance. 

Data from animal models lend insight into the role that small airway disease plays in the pathophysiology of BPD. Mouse models of BPD reveal that in a two-hit (antenatal LPS exposure and hyperoxia) model of BPD, structural airway changes are apparent. Airways in BPD mice demonstrate increased epithelial thickness, increased sub-epithelial collagen, and higher levels of alpha-smooth muscle actin expression [69,70,71]. The Airways of affected mice also exhibit increased contraction when exposed to broncho constrictors [69]. 

Structurally immature distal airways are also susceptible to dynamic obstruction in premature infants secondary to prolonged intubation and positive-pressure ventilation. Changes in neighboring lung tissue can destabilize the distal airways and result in decreased elastic recoil. These changes have been well-characterized in the adult lung disorder chronic obstructive pulmonary disease (COPD) [72]. Increasing evidence suggests similar changes occur in the lungs of premature infants affected by BPD [73]. In a prospective cohort study of 110 infants with severe BPD who underwent infant pulmonary function testing (iPFT) while admitted to the NICU, 91% were found to have a phenotype consistent with obstructive respiratory physiology, with 51% exhibiting a purely obstructive presentation and 40% displaying a mixed obstructive and restrictive pattern. Most neonates with evidence of obstructive physiology demonstrated improved airflow dynamics with bronchodilator treatment [74]. In this cohort, there were several factors noted to be associated with bronchodilator responsiveness, including a lower FEV0.5 prebronchodilator and a greater hyperinflation index [75]. The pattern of obstruction has been shown to persist into childhood and adulthood and can be associated with a reduced gas diffusion capacity [76,77,78]. 

In school-age children, forced airflow measurements can serve as an indicator for the degree of small airway obstruction, with reduced forced expiratory volumes in 1 s (FEV_1_) demonstrative of obstructed air flow. Studies of lung function performed in older children with a history of preterm birth and BPD consistently reveal lower forced expiratory volumes, with a stronger association seen in those with a history of severe BPD [79,80]. Recent reports have also suggested that children affected by BPD have a gradual decline in lung function with age, raising concern that a diagnosis of BPD may confer an increased risk of developing early-onset chronic obstructive pulmonary disease [81]. Recently, Simpson and colleagues detected a decline in several spirometric values (FEV1, FEF25–75, and FEV1/FVC) in ex-preterm infants tested between 4 and 12 years of age. Persistent respiratory symptoms, abnormal chest computed tomography (CT), and younger gestational age at birth were associated with poorer testing indices [81]. Interestingly, airway obstruction is more prevalent and more pronounced in those preterm infants who were also small for gestational age, suggesting that this may be an independent risk factor for more significant small airway disease. 

Adult survivors of BPD have been demonstrated to have significantly more distal airway obstruction than both ex-preterm infants without BPD and healthy ex-full-term subjects. A recent meta-analysis of 11 studies looking at expiratory airflow indices in adult survivors of prematurity revealed reduced airflow and lower Z scores for all expiratory flow measures when comparing ex-preterm participants to controls. This finding corroborates those of individual studies that suggest preterm infants, or those with very low birth weight, are not able to achieve their full airway growth potential [82]. Factors that lead to chronic lung dysfunction associated with small airways disease in children affected by BPD may include chronic airways inflammation, oxidative stress, air trapping, emphysematous changes in the lung parenchyma, and disproportionate growth between lung size and airway caliber (referred to as ‘dysanaptic growth’) as a result of asynchronous increases in lung size compared to airways [29,83,84]. A population-based birth-cohort study assessed pulmonary function including spirometry pre- and post-bronchodilator (salbutamol) administration, whole body plethysmography, and tests of lung diffusion capacity at 35 years of age for ex-premature infants born at gestational age (GA) less than 28 weeks or with birth weight ≤1000 g and compared these to matched term-born controls [77]. The authors reported pre-bronchodilator FEV1/FVC below the lower limits of normal with partial reversibility following salbutamol administration. They also noted bronchial obstruction and reduced gas diffusion capacity. Similarly, abnormal spirometry and lung diffusion findings were reported in a separate cohort of 53-year-old ex-premature infants [85]. 

Imaging performed in BPD survivors further elucidates the pathophysiology of the small airways in BPD. High-resolution CT scans of human subjects reveal that peribronchial thickening, hypoattenuation, and emphysematous areas with linear or triangular atelectasis (all evidence of small airways disease) are present in a subset of ex-preterm infants with a history of BPD and that these signs tend to decrease with age [86,87,88]. Recently, two research groups demonstrated that structural anomalies on CT were associated with decreased FEV1 compared with that of healthy controls, suggesting a correlation between small airway imaging findings and obstructive lung disease on spirometry [88,89].

## 4. Conclusions

Parenchymal lung abnormalities in BPD are relatively well-characterized and highly studied. To date, there has been less emphasis placed on the role that airways disease plays in the pathophysiology of BPD. Applying greater focus on understanding the impact that airway abnormalities play in this disease should be considered to enhance prevention strategies and facilitate the development of more phenotype-specific management approaches, especially in the most severely affected infants. 

Further studies are necessary to identify modalities to support normal airway development after birth, to minimize airway injury, and to identify patients that will benefit from specific treatment strategies aimed at preventing and managing airways disease in neonates with BPD.

## Figures and Tables

**Figure 1 children-10-01127-f001:**
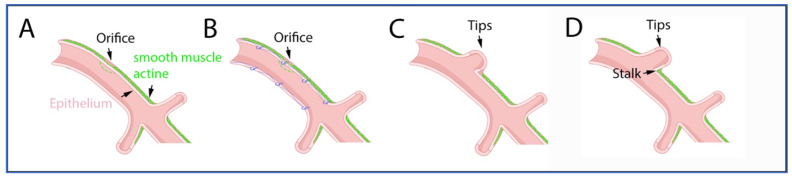
Branching morphogenesis. Embryonic lung with smooth muscle scells forming a sleeve around developing conducting airway epithelium (**A**). Interbranch lengths and timing of branching are thought to be partially determined by the presence of calcium waves (**B**) where lateral budding occurs at points of opening (orifice) in the smooth muscle coat (**C**). Sleeves of smooth muscle coats (stalk) extend toward the lateral budding of the epithelium (tips) while it protrudes to make new peripheral epithelial branches (**D**).

**Figure 2 children-10-01127-f002:**
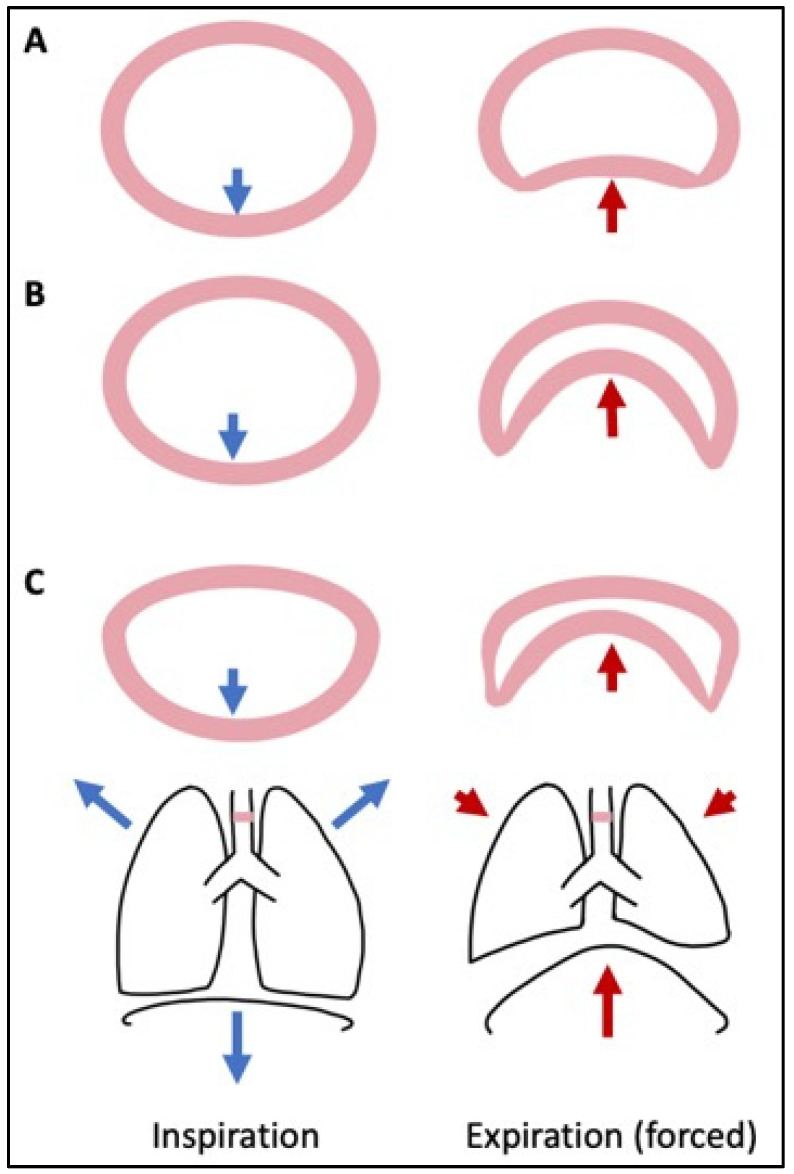
Effects of transmural forces on the neonatal airway. Transmural forces exhibited on the airways through diaphragmatic contraction on inspiration (blue arrows) and forceful exhalation (red arrows) in normal airways (**A**), tracheomalacia due to posterior trachealis instability (**B**), and a combination of extrinsic anterior compression of the trachea with posterior trachealis instability (**C**). Pink represents the area of cross-section highlighted in the cartoon (adapted from www.breathingnyc.com with permission, accessed on 16 January 2023).

**Table 1 children-10-01127-t001:** Anatomic airway locations, common pathologies and diagnostic studies to consider in neonates. Modified from Bush et al. 2019 [33].

Airway Region	Diagnosis	Etiology	Obstruction Type	Signs/Symptoms	Diagnostic Studies
Subglottis	Subglottic stenosis *	Acquired/Congenital	Fixed	Biphasic stridor/wheezing	FB, RB
	Subglottic hemangioma	Congenital	Fixed	Biphasic stridor/wheezing	FL, FB, RB
Trachea	Tracheoesophageal fistula	Congenital	Dynamic	Coughing with feeding, recurrent pneumonia	CXR, FB, or RB with methylene blue instillation
	Tracheomalacia *	Acquired	Dynamic	Expiratory wheezing	FB, MRI
	Tracheomegaly	Acquired	Dynamic	Wheezing	FB, RB, CT
	Tracheal compression (extrinsic) *	Acquired/Congenital	Dynamic or Fixed	Expiratory wheezing/Biphasic wheezing	FB, RB, CT
	Tracheal web	Congenital	Fixed	Biphasic wheezing	FB, RB
	Tracheal ring/stenosis	Acquired/Congenital	Fixed	Biphasic wheezing	RB, RB, CT
	Vascular ring, Pulmonary artery sling	Congenital	Fixed	Biphasic wheezing	FB, RB, CTA
	Tracheal bronchus	Congenital	Dynamic (regional)	Coughing, Retained secretions, Recurrent pneumonia	FB, RB, CT
Bronchi	Bronchomalacia *	Acquired	Dynamic	Expiratory wheezing	FB
	Bronchial stenosis	Acquired/Congenital	Fixed	Wheezing, air trapping	FB
	Bronchial compression (extrinsic) *	Acquired/Congenital	Dynamic or Fixed	Wheezing, air trapping	FB
	Airway granuloma	Acquired	Fixed	Wheezing, air trapping	FB, RB
Bronchioles	Obstructive airways disease *	Acquired	Fixed/Reversible ^	Wheezing, air trapping	PFT
	Atelectasis *	Acquired	Reversible	Reduced air entry, crackles	CXR

FL: flexible laryngoscopy; CT: computed tomography; CTA: computed tomography with angiography; FB: flexible bronchoscopy; RB: rigid bronchoscopy; CXR: chest radiograph; MRI: magnetic resonance imaging; PFT: pulmonary function testing (spirometry in school aged children, infant PFTs available at some centers). * Denotes pathology observed more frequently in bronchopulmonary dysplasia. ^ Reversibility in small airways disease in premature infants with bronchopulmonary dysplasia is not always evident.

## Data Availability

Not applicable.

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
