# Peer review of "Development and Disorders of the Airway in Bronchopulmonary Dysplasia"

_children, 2023, doi:10.3390/children10071127_

Round 1

Reviewer 1 Report

Summary:

In this review manuscript, the authors first described the lung development of humans and second, summarized the parenchymal lung abnormalities and their role in bronchopulmonary dysplasia.

General concerns:

1.     Please make “air flow” or “airflow” consistent in the manuscript.

2.     Please complete Author Contributions: Funding: Informed Consent Statement: Data Availability Statement: and Conflicts of Interest: at the end of the manuscript.

3.     Suggest adding a Table to describe the airway abnormalities in BPD.

Author Response

We are thankful for the time and the effort the reviewers spent evaluating our paper. We appreciate all the feedback provided and believe it made the paper easier to read and more scientifically robust.

Below are the point by point responses to the reviewer:

In this review manuscript, the authors first described the lung development of humans and second, summarized the parenchymal lung abnormalities and their role in bronchopulmonary dysplasia.

General concerns:

1. Please make “air flow” or “airflow” consistent in the manuscript.

  • We appreciate reviewer’s comment, and we made airflow consistent throughout the manuscript.

2. Please complete Author Contributions: Funding: Informed Consent Statement: Data Availability Statement: and Conflicts of Interest: at the end of the manuscript.

  • We apologize for this missing information, and we added author contributions, funding, informed consent statement, data availability and conflicts of interest at the end of the manuscript as suggested.

3. Suggest adding a Table to describe the airway abnormalities in BPD.

  • We really thank the reviewer for this request of clarification. We modified the table n.1, added small airway pathologies and identify with an asterix (*) the pathologies observed more frequently in bronchopulmonary dysplasia

Reviewer 2 Report

Dear author,

I just read your manuscript, in which you summarized the development of the airway as well as the large and small airway diseases that may develop in neonates with BPD.

It is a well written manuscript. I have only two minor comments to make.

#1 Figure 1 must be above its caption.

#2 Shouldn't lines 167-171 be in the Figure 2 caption?

#3 I didn't realized any distinction between BPD and new BPD throughout the text. Shouldn' t there be one? The pathophysiology is a bit different.

I find highly educative the way you presented the current literature.

The Table you summarized the different diseases is also very useful.

It is a well-written manuscript with sound use of the English language.

Minor editing is required.

Author Response

We are thankful for the time and the effort the reviewers spent evaluating our paper. We appreciate all the feedback provided and believe it made the paper easier to read and more scientifically robust.

Below are the point by point responses to the reviewer:

Dear author,

I just read your manuscript, in which you summarized the development of the airway as well as the large and small airway diseases that may develop in neonates with BPD.

It is a well written manuscript. I have only two minor comments to make.

#1 Figure 1 must be above its caption.

  • We thank the reviewer for pointing this out and we moved the figure 1 above its caption as suggested.          

#2 Shouldn't lines 167-171 be in the Figure 2 caption?                                                                                     

  • We appreciate the suggestion and add the following sentence in the figure 2 caption: "While there are no specific therapies available to treat tracheomalacia, non-invasive continuous positive airway pressure (CPAP) support can be used as treatment since it is providing an airway distending pressure, it helps reducing airways collapse and its resistance, facilitating exhalation and preventing hyperinflation".

#3 I didn't realized any distinction between BPD and new BPD throughout the text. Shouldn' t there be one? The pathophysiology is a bit different.

  • We thank the reviewer for this very important comment. We added the following sentence in the manuscript ( line 9-12): In its current form, often referred to as the ‘new BPD’ the pathophysiology is predominantly one of arrested or impaired development, which is in contrast to the features of lung injury that were the hallmark of the ‘old BPD’ phenotype.

I find highly educative the way you presented the current literature.

The Table you summarized the different diseases is also very useful.
